# Modeling and Simulation of a Novel Low-Speed High-Torque Permanent Magnet Synchronous Motor with Asymmetric Stator Slots

Shaokai Kou [1,2,3], Ziming Kou [1,2,3,4,*], Juan Wu [1,2,3] and Yandong Wang [1,2,3]

1 College of Mechanical and Vehicle Engineering, Taiyuan University of Technology, Taiyuan 030024, China
2 National-Local Joint Engineering Laboratory of Mining Fluid Control, Taiyuan 030024, China
3 Shanxi Province Engineering Technology Research Center for Mine Fluid Control, Taiyuan 030024, China
4 School of Mechanical Engineering, Anhui University of Science and Technology, Huainan 232001, China
* Correspondence: zmkou@163.com

**Abstract:** Focusing on the unstable electromagnetic performance of an air gap magnetic field caused by torque ripple and harmonic interference of a multi-slot and multi-pole low-speed high-torque permanent magnet synchronous motor (LHPMSM), an asymmetric stator slot is proposed to improve the comprehensive electromagnetic performance of the motor. Moreover, this paper develops an exact analytical model which predicts the magnetic field distribution based on Laplace's and Poisson's equations. The stator slot asymmetry is introduced into the model and solved by the method of separating variables. Taking a 40p168s LHPMSM as an example, numerical results of the no-load flux density field distributions are obtained by the finite element method (FEM) and employed to validate the analytical model. The influence of stator slot asymmetric structure on electromagnetic characteristics is subsequently analyzed. The results show that, compared with the semi-closed slot model, the asymmetric slot has better torque characteristics, and the electromagnetic characteristics of the motor can be significantly improved by optimizing the stator slot asymmetry. Finally, a prototype is manufactured and tested to validate the theoretical analysis.

**Keywords:** low-speed high-torque permanent magnet synchronous motor (LHPMSM); asymmetric slot; analytical model; slot asymmetry; electromagnetic performance

## 1. Introduction

The low-speed, high-torque permanent magnet synchronous motor (LHPMSM) has the advantage of low energy consumption, high efficiency, high power density, etc. It has been widely used in industrial applications such as mine transportation and mining, wind power generation, and many other fields [1–5]. However, the LHPMSM usually has the characteristics of multiple slots and multiple poles, large volume, and low installation accuracy. Moreover, the drive system employs the external motor and transmits torque through the output shaft, aggravating friction loss and increasing space volume. The commonly used transmission type of the motor is the external direct-drive permanent magnet motor or magnetic gear, with sufficient installation space. However, it is difficult to employ for some applications in limited space, such as fully mechanized mining faces and roadways in the underground mine.

This paper proposes a novel integrated LHPMSM which employs the transmission structure without a reducer, coupling, or high-speed shaft. Thus, the structure substantially reduces the total volume and assembly cost of the mechanical transmission system. However, its rotor bears the external load, which causes torque ripples. Therefore, it is necessary to improve the electromagnetic performance of the motor through structural design, especially for the stator and rotor.

Some research on improving the electromagnetic performance of the PMSM has been carried out. From the perspective of optimizing the structure of PMs, they mainly employ the segmented inclined magnetic pole [6,7], the segmented permanent magnet [8], the eccentric magnetic pole [9], the unequal-thickness magnetic pole [10,11], the combined magnetic pole [12], different pole arc coefficients [13,14], etc. However, for the multi-pole LHPMSM, the complex shape of permanent magnets greatly increases the production cost, with a large amount of material waste. Therefore, the inner arc eccentric PM described in [9] is adopted. The eccentricity of the PM is equal to the maximum magnetization thickness, which not only has low manufacturing cost, but also avoids material waste.

Optimizing the structure of stator slots is another effective technique to improve the electromagnetic performance of the LHPMSM. In [15], an eccentric stator tooth was proposed, and the influence of eccentricity on the cogging torque was analyzed; the results showed that compared with the non-eccentric structure, the eccentric structure had a more significant weakening effect on cogging torque. In [16], a novel arc-shaped tooth top stator slot structure for a multi-slot permanent magnet synchronous motor is proposed, the coupling relationship between tooth height and tooth top arc radius is established, and the tooth top circular arc radius value with the best electromagnetic characteristics (including cogging torque, torque ripple, air gap magnetic flux density harmonic, average electromagnetic torque) is obtained through a multi-objective optimization design method. In [17], a novel multi-tooth structure of two tooth permanent magnet brushless DC motor was proposed and investigated by 2D and 3D FEM models; the results revealed that the cogging torque of this structure accounts for only about 5% of the electromagnetic torque. In [18], a single-layer slot and double-layer slot combined with the stator core model for a multi-slot permanent magnet synchronous motor was proposed, in which the single-layer slot is composed of two asymmetric stator slots and three semi-closed slots. The influence of the winding factor on the harmonic of the magnetomotive force (MMF) was analyzed; the result shows that the MMF harmonics can be effectively suppressed and the output torque of the motor can be improved by properly selecting the winding factor ratio of the two slots. In [19], three optimization methods of stator slot parameters against a semi-closed slot, including changing the stator tooth width, unequal tooth width, and unequal slot width, are adopted to analyze the harmonic content of the torque ripple and vibration amplitude of electromagnetic force. The results show that the method of unequal tooth widths has the best reduction effect on the electromagnetic vibration and torque ripple. In [20], a slot-opening shift method was presented to reduce the cogging torque, and the calculation method for the shifting angle was expressed. In [21], a skewed unequal width adjacent stator tooth structure was proposed to weaken the 6th torque ripple harmonic of stator teeth caused by synchronous inductance difference; the axial skewed segment technology of PM was combined to point out that the 6th, 12th, and 24th cogging torque harmonics can be eliminated when the electrical angle of the half harmonic period is an integer multiple of the skew angle of adjacent PMs. Moreover, in [22,23], the Fourier decomposition coefficient of the cogging torque with an unequal tooth width and slot size of the stator was deduced, and its specific relationship with the adjacent stator tooth width and slot size was obtained. In [24], the effects of the slotted tooth, step tooth, and eccentric tooth on improving torque characteristics were compared by optimizing the slotting depth, step width, and eccentricity, respectively. Although the stepped tooth had better performance in reducing torque ripple and cogging torque than the slotted tooth, its average torque was the lowest. The average torque will be improved by increasing the number of steps. However, the harmonic content of the flux density was more severe. Therefore, the slotted tooth was more suitable for improving the torque density.

In this paper, an asymmetric slot structure is proposed, in which a V-shaped slot is opened on one side of the stator slot to reduce the edge flux linkage. On the other side, the semi-closed slot is retained to facilitate improving the average torque by optimizing the slot width. However, the complexity of the stator slot structure will inevitably lead to difficulty in magnetic field modeling, and it is challenging to obtain an accurate analytical

model by using the traditional subdomain (SD) model method. The exact SD model is an elegant way to analytically determine magnetic fields in electrical machines. This paper simplifies irregular areas in the stator slot as a sector annular. The asymmetry of the slot opening and slot body is defined to derive the asymmetric exact SD model.

The rest of this paper is organized as follows. Section 2 introduces the structure of the LHPMSM, as well as the asymmetric slot structure and its simplified model. Sections 3 and 4 describe the process of asymmetric SD modeling, including slot-opening asymmetric SD and slot body asymmetric SD. In Section 5, the FEA models of the semi-closed slot and asymmetric slot are introduced and simulated. Moreover, the effect of the slot opening asymmetry and slot body asymmetry on the electromagnetic performance is analyzed in Section 6. An LHPMSM prototype with a 40-pole/168-slot is manufactured and tested in Section 7. Conclusions are drawn in Section 8.

## 2. Structure and Analysis Model

### 2.1. Structure of LHPMSM

In this research, a 168-slot/40-pole LHPMSM is introduced to verify the asymmetric exact SD model. Figure 1 shows the 3D structure of the LHPMSM, which is a permanent magnet external-rotor mine hoist. In the whole configuration, the main shaft and stator are fixed by a flange; a hexagonal anti-torsion structure is adopted on both sides of the main shaft. Figure 2 shows the PMs adhered to the inner surface of the drum and fixed with circumferential and axial non-magnetic blocks. Moreover, the steel wire ropes (load) are wound on the outer surface of the drum. Figure 3 shows the configuration of the asymmetric stator slot.

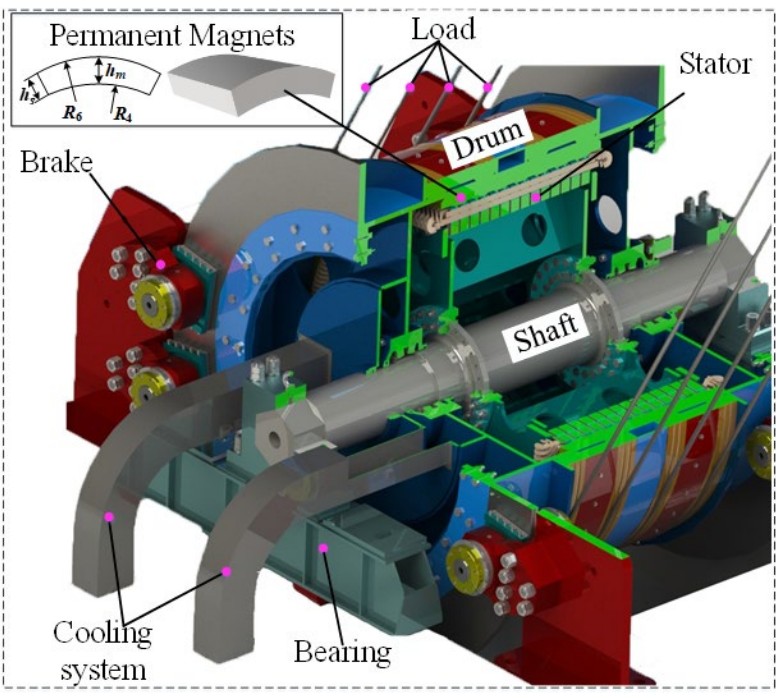

**Figure 1.** Three-dimensional structure of the LHPMSM.

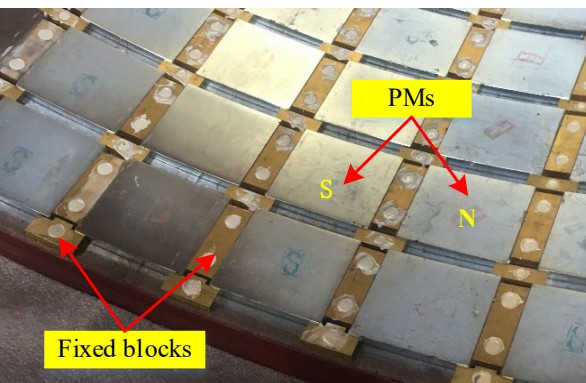

**Figure 2.** Configuration of the rotor.

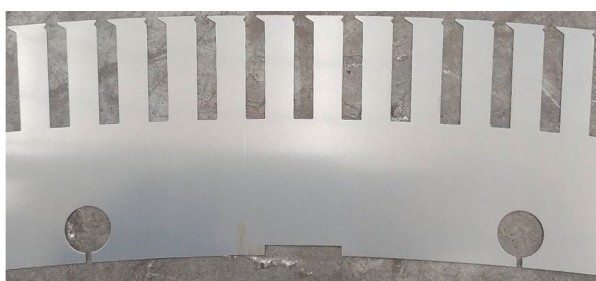

**Figure 3.** Configuration of the stator slot.

*2.2. Analysis Model*

Figure 4a shows the single asymmetric stator slot parametric model. The slot opening is the asymmetric structure in which the arc length from the center line of the slot opening to the slot body within the slot is unequal, and the slot opening angle is $\beta$. Moreover, one side of the tooth tip is collinear with the slot body on the same side. The angles from the center line of the slot opening to the slot body on both sides are $\beta/2$ and $\beta 1$, respectively. On one side of the slot body, the structure can be equivalent to opening a U-shaped groove, where l is the groove depth. The other side retains the unilateral structure of the semi-closed slot. To facilitate modeling, we simplified the original model so that the irregular region in the slot is simplified as the sector annular area. The angle from the bottom edge line on one side of the area to the slot body is $\alpha$, and the slot-pitch angle is $\delta$. Then, we can define the following geometric relationships:

(1) The asymmetry of the slot opening: the deviation degree of the center line of slot opening relative to the center line of stator slot, which is defined as $\lambda$, as shown in Equation (1). It should be noted that when $\beta 1 < \delta/2$, the slotting effect has a serious impact on the magnetic flux density. Therefore, this situation is ignored during the analysis.

$$\lambda = \frac{2\beta_1}{\delta} = \frac{2\delta - \beta}{\delta} \left( \frac{\delta}{2} \le \beta < \delta,\ 1.5 \le \lambda < 2 \right) \tag{1}$$

(2) The asymmetry of the inside slot: the deviation degree of the center line of the asymmetric region within the slot relative to the center line of the stator slot, which is defined as

$$\xi = \frac{\alpha + \delta}{\delta} \quad (0 < a < \delta,\ 1 < \xi < 2) \tag{2}$$

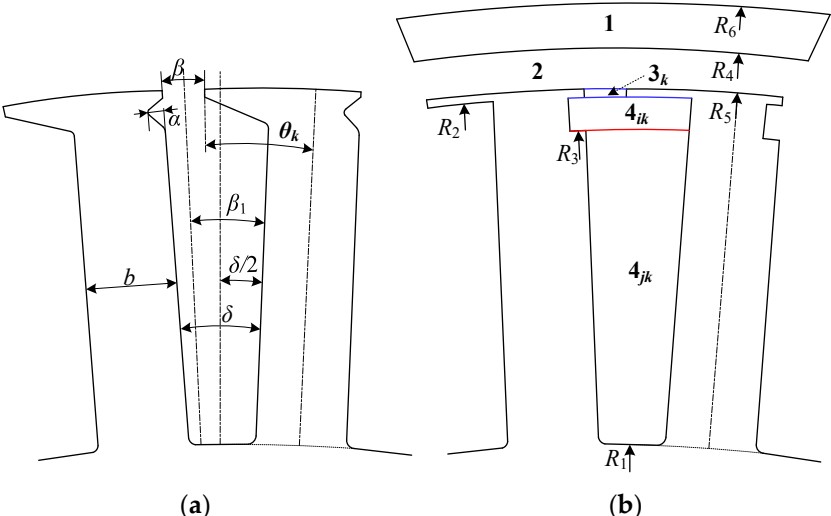

**Figure 4.** Single asymmetric stator slot model. (**a**) Original parametric model; (**b**) simplified model.

## 3. Analytical Solution of Magnetic Field

In this paper, the following assumptions are made to enable and simplify the analytical solution: (1) the PMs adopt radial magnetization; (2) the end effect is neglected; (3) the permeability of stator iron is infinite; (4) the relative permeability of PM is equal to 1.

Figure 4a shows the 1/8 geometry of LHPMSM with an asymmetric slot. Figure 4b shows the solution domain of the motor, which is divided into five subdomains: PMs (Region 1), air gap (Region 2), asymmetric slot opening (Region 3k), asymmetric slot body (Region 4ik), and symmetrical slot body (Region 4jk), with $k = 1, 2, \ldots, Q$, and $Q$ being the number of slots.

The parameters in the model are radius of slot bottom, $R_1$, inner radius of slot opening, $R_2$, outer radius of symmetrical slot body, $R_3$, inner radius of PMs, $R_4$, outer radius of slot opening, $R_5$, outer radius of PMs, $R_6$, and the initial position of kth slot, $\theta_k$. $\theta_k$ and $\delta$ can be expressed by $\lambda$ as

$$\beta = (2 - \lambda)\delta \tag{3}$$

$$\theta_k = \frac{k\pi}{Q} + \left(\frac{\delta}{2} - \beta\right) = \frac{k\pi}{Q} + \left(\lambda - \frac{3}{2}\right)\delta \tag{4}$$

The $s$th subdomain magnetic vector potential $A_s$ satisfies the governing function as

$$\nabla^2 A_s = \begin{cases} 0 & s = 2, 3k, 4ik \\ \frac{\mu_0}{r} \cdot \frac{\partial M_r}{\partial \theta} & s = 1 \\ -\mu_0 J & s = 4jk1, 4jk2 \end{cases} \tag{5}$$

where $\mu_0$, $M_r$ and $J$ are the permeability of vacuum, radial component of the magnetization vector in the PM subdomain, and current density in the slot, respectively.

The general solutions of the vector potential $A_s$ can be expressed as follows [25]:

$$
\begin{aligned}
A_s(r, \theta) = {}& A_{s0} + B_{s0} \ln r \\
& + \sum_{n=1}^{\infty} (A_{sn} f_{sa}(r) + B_{sn} f_{sb}(r)) \cdot \cos(p_s \theta_s) \\
& + \sum_{n=1}^{\infty} (C_{sn} f_{sa}(r) + D_{sn} f_{sb}(r)) \cdot \sin(p_s \theta_s) + A_{sp}
\end{aligned} \tag{6}
$$

where $A_{s0}$, $B_{s0}$, $A_{sn}$, $B_{sn}$, $C_{sn}$ and $D_{sn}$ are integral constants; $f_{sa}$ and $f_{sb}$ are functions of the r; $\theta_s$ is the angle of $s$th subdomain, $\theta_s = \theta - \theta_{s0}$, $\theta_{s0}$ is the initial phase angle of $s$th subdomain; $\eta_s$ is the Fourier series period. The expressions of $f_{sa}$, $f_{sb}$ and $p_s$ are

$$f_{sa}(r) = \left(\frac{r}{R_{sO}}\right)^{\frac{2n\pi}{\eta_s}}, f_{sb}(r) = \left(\frac{r}{R_{sI}}\right)^{-\frac{2n\pi}{\eta_s}} \text{ and } p_s = \frac{2n\pi}{\eta_s}$$

where $R_{sO}$ and $R_{sI}$ are the outer radius and inner radius of $s$th subdomains, respectively.

### 3.1. Field Solutions in Asymmetric Slot Opening

As shown in Figure 4a, the span angle of $k$th asymmetric slot opening ranges from $\theta_k$ to $\theta_k + \beta$. The Laplace equation in polar coordinates of the asymmetric slot opening subdomain is

$$\begin{cases} \frac{\partial^2 A_{3k}}{\partial r^2} + \frac{1}{r}\frac{\partial A_{3k}}{\partial r} + \frac{1}{r^2}\frac{\partial^2 A_{3k}}{\partial \theta^2} = 0 \\ R_2 \le r \le R_5 , \; \theta_k \le \theta \le \theta_k + \beta \end{cases} \tag{7}$$

As can be seen from Figure 5c,d, the boundary conditions of the $k$th slot opening subdomain write

$$\left.\frac{\partial A_{3k}}{\partial \theta}\right|_{\theta=\theta_k} = 0, \; \left.\frac{\partial A_{3k}}{\partial \theta}\right|_{\theta=\theta_k+\beta} = 0 \tag{8}$$

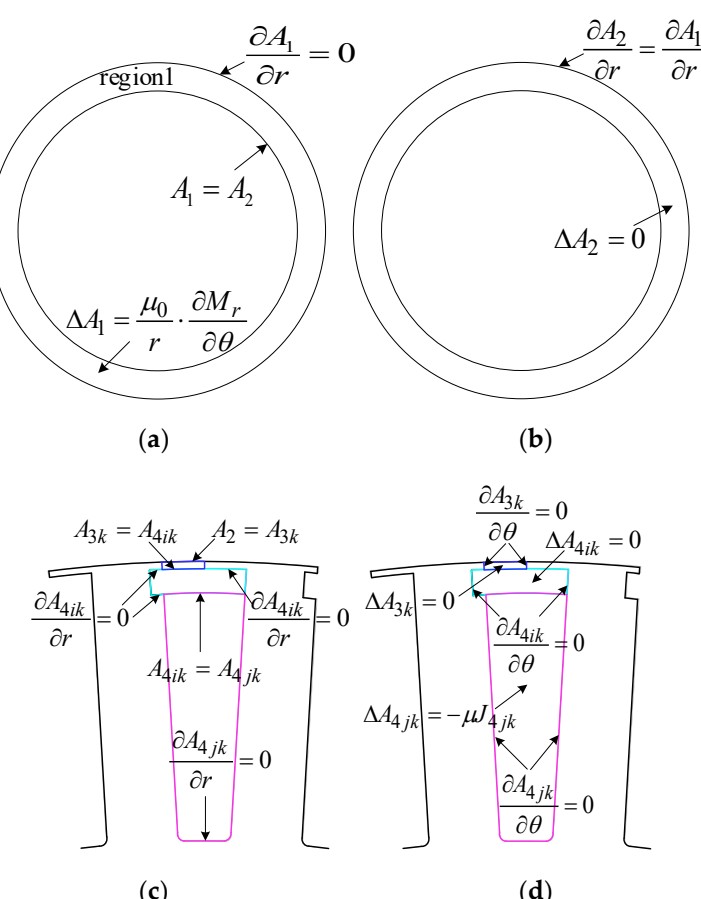

**Figure 5.** Boundary conditions of each subdomain. (**a**) PMs; (**b**) air gap; (**c**) radial asymmetric slot opening and slot body; (**d**) circumferential asymmetric slot opening and slot body.

The continuity of the radial component of the flux density leads to

$$A_{3k}(r,\theta)|_{r=R_2} = A_{4ik}(r, \theta)|_{r=R_2} \tag{9}$$

$$A_{3k}(r,\theta)|_{r=R_5} = A_2(r,\theta)|_{r=R_5} \tag{10}$$

Taking into account (6) and (7) and the boundary conditions shown in Equation (8), the general solution of vector potential $A_{3k}$ in the asymmetric slot opening subdomain can be derived as

$$A_{3k}(r,\theta) = A_{3k} + B_{3k}\ln r + \sum_{m=1}^{\infty}\left(A_{3km}f_{3ka}(r) + B_{3km}f_{3kb}(r)\right)\cdot\cos(p_{3k}\theta_{3k}) \tag{11}$$

where

$$f_{3ka}(r) = \left(\frac{r}{R_5}\right)^{\frac{m\pi}{(2-\lambda)\delta}}, f_{3kb}(r) = \left(\frac{r}{R_2}\right)^{-\frac{m\pi}{(2-\lambda)\delta}}, \theta_{3k} = \theta - \theta_k \text{ and } p_{3k} = \frac{m\pi}{(2-\lambda)\delta}$$

The constants $A_{3k}$, $B_{3k}$, $A_{3kn}$, and $B_{3kn}$ are determined using Fourier series expansion as

$$A_{3k} + B_{3k}\ln R_2 = \frac{1}{(2-\lambda)\delta}\int_{\theta_k}^{\theta_k+(2-\lambda)\delta} A_{4ik}(R_2,\theta)d\theta \tag{12}$$

$$A_{3k} + B_{3k}\ln R_5 = \frac{1}{(2-\lambda)\delta}\int_{\theta_k}^{\theta_k+(2-\lambda)\delta} A_2(R_5,\theta)d\theta \tag{13}$$

$$A_{3km} = \frac{2}{(2-\lambda)\delta}\int_{\theta_k}^{\theta_k+(2-\lambda)\delta} A_{4ik}(R_2,\theta)\cos\frac{m\pi}{(2-\lambda)\delta}(\theta-\theta_k)d\theta \tag{14}$$

$$B_{3km} = \frac{2}{(2-\lambda)\delta}\int_{\theta_k}^{\theta_k+(2-\lambda)\delta} A_2(R_5,\theta)\cos\frac{m\pi}{(2-\lambda)\delta}(\theta-\theta_k)d\theta \tag{15}$$

### 3.2. Field Solutions in Asymmetric Slot Body

As shown in Figure 4b, it can be seen that the span angle of the $k$th asymmetric slot body ranges from $\theta_k - (\lambda - 1)\delta$ to $\theta_k + \delta\xi - (\lambda - 1)\delta$. The Laplace equation in polar coordinates of the asymmetric slot body subdomain is

$$\begin{cases} \dfrac{\partial^2 A_{4ik}}{\partial r^2} + \dfrac{1}{r}\dfrac{\partial A_{4ik}}{\partial r} + \dfrac{1}{r^2}\dfrac{\partial^2 A_{4ik}}{\partial\theta^2} = 0 \\ R_3 \le r \le R_2, \theta_k - (\lambda-1)\delta \le \theta \le \theta_k + \delta\xi - (\lambda-1)\delta \end{cases} \tag{16}$$

As can be seen in Figure 5c,d, the boundary conditions of the $k$th asymmetric slot body subdomain are

$$\frac{\partial A_{4ik}}{\partial\theta}\bigg|_{\theta=\theta_k-(\lambda-1)\delta} = 0, \quad \frac{\partial A_{4ik}}{\partial\theta}\bigg|_{\theta=\theta_k+\delta\xi-(\lambda-1)\delta} = 0 \tag{17}$$

The continuity of the radial component of the flux density leads to

$$A_{4ik}(r,\theta)|_{r=R_3} = A_{4jk1}(r,\theta)\bigg|_{r=R_3} \tag{18}$$

Similarly, the general solution of vector potential $A_{3k}$ in the asymmetric slot body subdomain can be derived as

$$A_{4ik}(r,\theta) = A_{4ik} + B_{4ik}\ln r + \sum_{u=1}^{\infty}\left(A_{4iku}f_{4ikua}(r) + B_{4iku}f_{4ikub}(r)\right)\cdot\cos(p_{4iku}\theta_{4ik}) \tag{19}$$

where

$$f_{4ikva}(r) = \left(\frac{r}{R_2}\right)^{\frac{u\pi}{\delta\xi}}, f_{4ikb}(r) = \left(\frac{r}{R_3}\right)^{-\frac{u\pi}{\delta\xi}}, \theta_{4ik} = \theta - \theta_k + (\lambda-1)\delta \text{ and } p_{4iku} = \frac{u\pi}{\delta\xi}$$

The constants $A_{4ik}$, $B_{4ik}$, $A_{4ikn}$, and $B_{4ikn}$ are determined using Fourier series expansion:

$$A_{4ik} + B_{4ik} \ln R_2 = \frac{1}{\delta\xi} \int_{\theta_k-(\lambda-1)\delta}^{\theta_k+\delta\xi-(\lambda-1)\delta} A_{3k}(R_2,\theta)d\theta \tag{20}$$

$$A_{4ik} + B_{4ik} \ln R_3 = \frac{1}{\delta\xi} \int_{\theta_k-(\lambda-1)\delta}^{\theta_k+\delta\xi-(\lambda-1)\delta} A_{4jk1}(R_3,\theta)d\theta \tag{21}$$

$$A_{4iku} = \frac{2}{\delta\xi} \int_{\theta_k-(\lambda-1)\delta}^{\theta_k+\delta\xi-(\lambda-1)\delta} A_{3k}(R_2,\theta)\cos\frac{u\pi}{\delta\xi}(\theta-\theta_k+(\lambda-1)\delta)d\theta \tag{22}$$

$$B_{4iku} = \frac{2}{\delta\xi} \int_{\theta_k-(\lambda-1)\delta}^{\theta_k+\delta\xi-(\lambda-1)\delta} A_{4jk1}(R_3,\theta)\cos\frac{u\pi}{\delta\xi}(\theta-\theta_k+(\lambda-1)\delta)d\theta \tag{23}$$

### 3.3. Cogging Torque Calculation

In the polar coordinate system, the radial and tangential flux density distribution in the air gap is given by

$$B_{r2}(r,\theta) = \frac{1}{r}\frac{\partial A_2(r,\theta)}{\partial\theta}, B_{\theta2}(r,\theta) = -\frac{\partial A_2(r,\theta)}{\partial r} \tag{24}$$

where $B_{r2}$ and $B_{\theta2}$ are the radial and tangential components of the air gap flux density at $r$, respectively.

According to the Maxwell stress tensor, a circle of radius rag in the air-gap subdomain is taken as the integration path. The electromagnetic torque is expressed as follows:

$$T_c = \frac{L_{ef}r_{ag}^2}{\mu_0} \int_0^{2\pi} B_{r2}(r_{ag},\theta)\cdot B_{\theta2}(r_{ag},\theta)d\theta \tag{25}$$

where $L_{ef}$ is the stack length and $r_{ag}$ is the radius of the air-gap subdomain.

### 3.4. Field Solutions in the Rest Subdomains

According to Equation (5), the magnetic vector potential expressions of the permanent magnet subdomain, the air gap subdomain, and the symmetrical slot body subdomain are given by

$$\text{region1}: \begin{cases} \frac{\partial^2 A_1(r,\theta)}{\partial r^2} + \frac{1}{r}\frac{\partial A_1(r,\theta)}{\partial r} + \frac{1}{r^2}\frac{\partial^2 A_1(r,\theta)}{\partial\theta^2} = \frac{\mu_0}{r}\cdot\frac{\partial M_r}{\partial\theta} \\ R_4 \le r \le R_6, \ 0 \le \theta \le 2\pi \end{cases} \tag{26}$$

$$\text{region2}: \begin{cases} \frac{\partial^2 A_2(r,\theta)}{\partial r^2} + \frac{1}{r}\frac{\partial A_2(r,\theta)}{\partial r} + \frac{1}{r^2}\frac{\partial^2 A_2(r,\theta)}{\partial\theta^2} = 0 \\ R_5 \le r \le R_4, \ 0 \le \theta \le 2\pi \end{cases} \tag{27}$$

$$\text{region4}jk1: \begin{cases} \frac{\partial^2 A_{4jk1}(r,\theta)}{\partial r^2} + \frac{1}{r}\frac{\partial A_{4jk1}(r,\theta)}{\partial r} + \frac{1}{r^2}\frac{\partial^2 A_{4jk2}(r,\theta)}{\partial\theta^2} = -\mu_0 J_{4jk1} \\ R_{4jk12} \le r \le R_3, \ \theta_k-(\lambda-1)\delta \le \theta \le \theta_k+(2-\lambda)\delta \end{cases} \tag{28}$$

$$\text{region4}jk2: \begin{cases} \frac{\partial^2 A_{4jk2}(r,\theta)}{\partial r^2} + \frac{1}{r}\frac{\partial A_{4jk2}(r,\theta)}{\partial r} + \frac{1}{r^2}\frac{\partial^2 A_{4jk2}(r,\theta)}{\partial\theta^2} = -\mu_0 J_{4jk2} \\ R_1 \le r \le R_{4jk12}, \ \theta_k-(\lambda-1)\delta \le \theta \le \theta_k+(2-\lambda)\delta \end{cases} \tag{29}$$

where $J_{4jk1}$ and $J_{4jk2}$ is the current density.

As shown in Figure 5, the associated boundary conditions in PMs, air-gap, and slot subdomains are

$$\text{region1}: \left.\frac{\partial A_1}{\partial r}\right|_{r=R_6} = 0 \tag{30}$$

$$A_1(r,\theta)|_{r=R_4} = A_2(r,\theta)|_{r=R_4} \tag{31}$$

$$H_1(r,\theta)|_{r=R_4} = H_2(r,\theta)|_{r=R_4} \tag{32}$$

$$\text{region2}: \left. \frac{\partial A_2}{\partial r} \right|_{r=R_5} = \begin{cases} \left. \frac{\partial A_{3k}}{\partial r} \right|_{r=R_5} & \theta_k \leq \theta \leq \theta_k + \beta \\ 0 & \text{elsewhere} \end{cases} \tag{33}$$

$$A_2(r,\theta)|_{r=R_5} = \begin{cases} A_{3k}(r,\theta)|_{r=R_5} & \theta_k \leq \theta \leq \theta_k + \beta \\ 0 & \text{elsewhere} \end{cases} \tag{34}$$

$$\text{region4}jk1: \left. \frac{\partial A_{4jk1}}{\partial r} \right|_{r=R_3} = \begin{cases} \left. \frac{\partial A_{4ik}}{\partial r} \right|_{r=R_3} & \theta_k - (\lambda-1)\delta \leq \theta \leq \theta_k + (2-\lambda)\delta \\ 0 & \text{elsewhere} \end{cases} \tag{35}$$

$$A_{4jk1}(r,\theta) \Big|_{r=R_3} = A_{4ik}(r,\theta) \Big|_{r=R_3}, \theta_k - (\lambda-1)\delta \leq \theta \leq \theta_k + (2-\lambda)\delta \tag{36}$$

$$\left. \frac{\partial A_{4jk1}}{\partial r} \right|_{r=R_{4jk13}} = \left. \frac{\partial A_{4jk2}}{\partial r} \right|_{r=R_{4jk13}} \tag{37}$$

$$A_{4jk1}(r,\theta) \Big|_{r=R_{4jk13}} = A_{4jk2}(r,\theta) \Big|_{r=R_{4jk13}} \tag{38}$$

$$\left. \frac{\partial A_{4jk1}}{\partial \theta} \right|_{\theta=\theta_k-(\lambda-1)\delta} = 0, \quad \left. \frac{\partial A_{4jk1}}{\partial \theta} \right|_{\theta=\theta_k+\delta\xi-(\lambda-1)\delta} = 0 \tag{39}$$

$$\text{region4}jk2: \left. \frac{\partial A_{4jk1}}{\partial r} \right|_{r=R_1} = 0 \tag{40}$$

$$\left. \frac{\partial A_{4jk2}}{\partial \theta} \right|_{\theta=\theta_k-(\lambda-1)\delta} = 0, \quad \left. \frac{\partial A_{4jk2}}{\partial \theta} \right|_{\theta=\theta_k+\delta\xi-(\lambda-1)\delta} = 0 \tag{41}$$

where $R_{4jk13} = \sqrt{(R_1^2 + R_3^2)/2}$.

The general solution of vector potential $A_1$, can be derived as

$$
\begin{aligned}
A_1(r,\theta) \quad &= \sum_{n=1}^{\infty} \left( A_{1n}\left(\frac{r}{R_6}\right)^n + B_{1n}\left(\frac{r}{R_4}\right)^{-n} \right) \cdot \cos n\theta \\
&+ \sum_{n=1}^{\infty} \left( C_{1n}\left(\frac{r}{R_6}\right)^n + D_{1n}\left(\frac{r}{R_4}\right)^{-n} \right) \cdot \sin n\theta + A_{1p} \\
A_{1p}(r,\theta) &= \begin{cases} \sum_{n=1}^{\infty} \left[ \frac{\mu_0 n M_{rn} r}{n^2-1} (-\cos(n\theta) + \sin(n\theta)) \right] n \neq 1 \\ \sum_{n=1}^{\infty} \left[ \frac{\mu_0 M_{rn} r \ln r}{2} (\cos(n\theta) - \sin(n\theta)) \right] n = 1 \end{cases}
\end{aligned}
\tag{42}
$$

where $M_{rn}$ is the radial component for remanent magnetization of PMs.

Taking into account the boundary conditions (30), the following equations can be obtained:

$$B_{1n} = \begin{cases} \left( A_{1n} - M_{rn} \cdot \frac{R_6 \mu_0}{n^2-1} \right) \cdot \left(\frac{R_4}{R_6}\right)^n & n \neq 1 \\ \left( A_{1n} + M_{rn} \frac{R_6(1+\ln R_6)\mu_0}{n(n^2-1)} \right) \cdot \left(\frac{R_4}{R_6}\right)^n & n = 1 \end{cases} \tag{43}$$

$$D_{1n} = \begin{cases} \left( C_{1n} + M_{rn} \cdot \frac{R_6 \mu_0}{n^2-1} \right) \cdot \left(\frac{R_4}{R_6}\right)^n & n \neq 1 \\ \left( C_{1n} - M_{rn} \frac{R_6(1+\ln R_6)\mu_0}{n(n^2-1)} \right) \cdot \left(\frac{R_4}{R_6}\right)^n & n = 1 \end{cases} \tag{44}$$

Then, according to Equations (42)–(44), the vector potential $A_1(r,\theta)$ can be rewritten as

$$
\begin{aligned}
A_1(r,\theta) \quad &= \sum_{n=1}^{\infty} [A_{1n} f_1(r) - M_{rn} f_{1rn}(r)] \cdot \cos n\theta \\
&+ \sum_{n=1}^{\infty} [C_{1n} f_1(r) + M_{rn} f_{1rn}(r)] \cdot \sin n\theta
\end{aligned}
\tag{45}
$$

where

$$f_1(r) = \left[ \left( \frac{r}{R_6} \right)^n + \left( \frac{R_4}{R_6} \right)^n \left( \frac{r}{R_4} \right)^{-n} \right]$$

$$f_{1rn}(r) = \begin{cases} \frac{\mu_0}{n^2-1} \left( R_6 \cdot \left( \frac{R_4}{R_6} \right)^n \left( \frac{r}{R_4} \right)^{-n} \right) + nr & n \neq 1 \\ -\frac{\mu_0}{2} \left( \frac{R_6(1+\ln R_6)}{n} \left( \frac{R_4}{R_6} \right)^n \left( \frac{r}{R_4} \right)^{-n} + r \ln r \right) & n = 1 \end{cases}$$

The general solution of vector potential $A_2$ can be derived as

$$\begin{aligned} A_2(r,\theta) &= \sum_{n=1}^{\infty} \left( A_{2n} \left( \frac{r}{R_4} \right)^n + B_{2n} \left( \frac{r}{R_5} \right)^{-n} \right) \cdot \cos n\theta \\ &+ \sum_{n=1}^{\infty} \left( C_{2n} \left( \frac{r}{R_4} \right)^n + D_{2n} \left( \frac{r}{R_5} \right)^{-n} \right) \cdot \sin n\theta \end{aligned} \tag{46}$$

As shown in Figure 6, the region 4*jk* adopts double-layer overlapping short-distance winding, with the upper and lower slot areas being equal. The general solution of vector potential $A_{4jk1}$ and $A_{4jk2}$ can be derived as

$$A_{4jk1}(r,\theta) = A_{4jk1} + \sum_{u=1}^{\infty} \left( A_{4jku1} f_{4jkua1}(r) + B_{4jku1} f_{4jkub1}(r) \right) \cdot \cos(p_{4jku1} \theta_{4jk1}) + A_{4jk1p}$$
$$f_{4jkua1}(r) = \left( \frac{r}{R_3} \right)^{\frac{u\pi}{\delta}}, f_{4jkub1}(r) = \left( \frac{r}{R_{4jk13}} \right)^{-\frac{u\pi}{\delta}} \tag{47}$$
$$\theta_{4jk1} = (\theta - \theta_k + (\lambda - 1)\delta)$$
$$p_{4jku1} = \frac{u\pi}{\delta}$$

$$l A_{4jk2}(r,\theta) = A_{4jk2} + \sum_{u=1}^{\infty} \left( A_{4jku2} f_{4jkua2}(r) + B_{4jku2} f_{4jkub2}(r) \right) \cdot \cos(p_{4jku2} \theta_{4jk2}) + A_{4jku2p}$$
$$f_{4jkua2}(r) = \left( \frac{r}{R_{4jk13}} \right)^{\frac{u\pi}{\delta}}, f_{4jkub2}(r) = \left( \frac{r}{R_1} \right)^{-\frac{u\pi}{\delta}} \tag{48}$$
$$\theta_{4jk2} = (\theta - \theta_k + (\lambda - 1)\delta)$$
$$p_{4jku2} = \frac{u\pi}{\delta}$$

$$A_{4jku1p} = \frac{1}{2}\mu_0 J_{4jk1}(R_{4jk13}^2 \ln r - \frac{1}{2}r^2) + \frac{1}{2}\mu_0 J_{4jk2}\left( R_1^2 - R_{4jk13}^2 \right) \ln r \tag{49}$$

$$A_{4jku2p} = \frac{1}{2}\mu_0 J_{4jk2}(R_1^2 \ln r - \frac{1}{2}r^2) \tag{50}$$

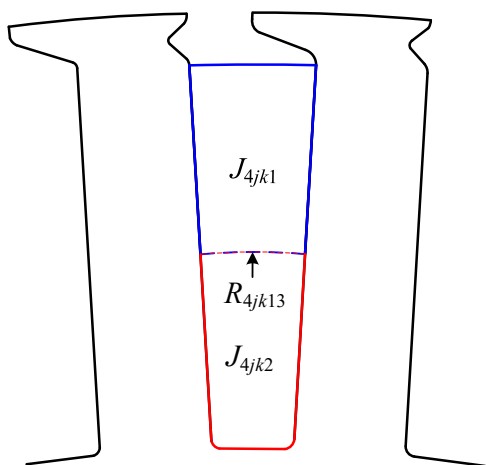

**Figure 6.** Winding configurations.

## 4. Integration Constants

In order to obtain the integration constants in the expressions of the vector potential, the interface conditions between subdomains need to be applied.

### 4.1. Interface between Region 1 and Region 2

According to Equations (31), (45) and (46), the following equations can be obtained:

$$2A_{1n}\left(\frac{R_4}{R_6}\right)^n - A_{2n} - B_{2n}\left(\frac{R_4}{R_5}\right)^{-n} = M_{rn}f_{1rn}(R_4) \tag{51}$$

$$2C_{1n}\left(\frac{R_4}{R_6}\right)^n - C_{2n} - D_{2n}\left(\frac{R_4}{R_5}\right)^{-n} = -M_{rn}f_{1rn}(R_4) \tag{52}$$

The interface condition (32) can be rewritten as

$$H_1(r,\theta)|_{r=R_4} = H_2(r,\theta)|_{r=R_4} \rightarrow \frac{1}{\mu_r}\cdot\frac{\partial A_1(r,\theta)}{\partial r}\bigg|_{r=R_4} = \frac{\partial A_2(r,\theta)}{\partial r}\bigg|_{r=R_4} \tag{53}$$

According to Equations (45), (46) and (53), the following equations can be obtained:

$$A_{2n} - B_{2n}\left(\frac{R_4}{R_5}\right)^{-n} = M_{rn}f_{2rn}(R_4) \tag{54}$$

$$C_{2n} - D_{2n}\left(\frac{R_4}{R_5}\right)^{-n} = -M_{rn}f_{2rn}(R_4) \tag{55}$$

where

$$f_{2rn}(R_4) = \begin{cases} \left[\frac{n}{R_4}\frac{\mu_0}{n^2-1}\left(R_6\cdot\left(\frac{R_4}{R_6}\right)^n\right) - n\right] & n \neq 1 \\ \left[-\frac{n}{R_4}\frac{\mu_0}{2}\frac{R_6(1+\ln R_6)}{n}\left(\frac{R_4}{R_6}\right)^n + \frac{\mu_0}{2}(1+\ln R_4)\right] & n = 1 \end{cases}$$

### 4.2. Interface between Region 2 and Region 3k

According to Equations (13), (15), (33) and (34), the following equation can be obtained:

$$\begin{aligned} A_{3k} + B_{3k}\ln R_5 \quad = \frac{1}{(2-\lambda)\delta}\Bigg[ &\sum_{n=1}^{\infty}\left(A_{2n}\left(\frac{R_5}{R_4}\right)^n + B_{2n}\right)\cdot l(n,k) \\ &+ \sum_{n=1}^{\infty}\left(C_{2n}\left(\frac{R_5}{R_4}\right)^n + D_{2n}\right)\cdot s(n,k)\Bigg] \end{aligned} \tag{56}$$

$$\begin{aligned} B_{3km} = \frac{2}{(2-\lambda)\delta}\Bigg\{ &\sum_{n=1}^{\infty}\left(A_{2n}\left(\frac{R_5}{R_4}\right)^n + B_{2n}\right)f(m,n,k) \\ &+ \sum_{n=1}^{\infty}\left(C_{2n}\left(\frac{R_5}{R_4}\right)^n + D_{2n}\right)g(m,n,k)\Bigg\} \end{aligned} \tag{57}$$

where

$$l_{2,3k}(n,k) = \int_{\theta_k}^{\theta_k+(2-\lambda)\delta}\cos n\theta d\theta$$

$$s_{2,3k}(n,k) = \int_{\theta_k}^{\theta_k+(2-\lambda)\delta}\sin n\theta d\theta$$

$$\gamma_{2,3k}(n,k) = \int_{\theta_k}^{\theta_k+(2-\lambda)\delta}\sin\left(\frac{m\pi}{(2-\lambda)\delta}(\theta-\theta_k)\right)d\theta$$

$$f_{2,3k}(m,n,k) = \int_{\theta_k}^{\theta_k+(2-\lambda)\delta}\cos n\theta\cdot\cos\left(\frac{m\pi}{(2-\lambda)\delta}(\theta-\theta_k)\right)d\theta$$

$$g_{2,3k}(m,n,k) = \int_{\theta_k}^{\theta_k+(2-\lambda)\delta}\sin n\theta\cdot\cos\left(\frac{m\pi}{(2-\lambda)\delta}(\theta-\theta_k)\right)d\theta$$

Then, according to Equation (10) or Equation (34), the following equation can be obtained:

$$A_{2n}\left(\frac{R_5}{R_4}\right)^n + B_{2n} = \sum_{m=1}^{\infty}\left[A_{3km} + B_{3km}\left(\frac{R_5}{R_2}\right)^{-\frac{m\pi}{(2-\lambda)\delta}}\right]\cdot f(m,n,k) \tag{58}$$

$$C_{2n}\left(\frac{R_5}{R_4}\right)^n + D_{2n} = \sum_{m=1}^{\infty}\left[A_{3km} + B_{3km}\left(\frac{R_5}{R_2}\right)^{-\frac{m\pi}{(2-\lambda)\delta}}\right]\cdot g(m,n,k) \tag{59}$$

### 4.3. Interface between Region 3k and Region 4ik

According to Equations (9), (11), (12) (14), (19), (20) and (22), the following equation can be obtained:

$$\begin{aligned}A_{3km} \quad &= \frac{2}{(2-\lambda)\delta}\Big[(A_{4ik} + B_{4ik}\ln R_2)l_{3k,4ik}(m,k)\\ &+ \sum_{u=1}^{\infty}\left(A_{4iku} + B_{4iku}\left(\frac{r}{R_2}\right)^{-\frac{u\pi}{\delta\xi}}\right)f_{3k,4ik}(u,m,k)\Big]\end{aligned} \tag{60}$$

$$A_{4iku} = \frac{2}{\delta\xi}\cdot\left[A_{3k} + B_{3k}\ln R_2 + \sum_{m=1}^{\infty}\left(A_{3km}\left(\frac{R_2}{R_5}\right)^{\frac{m\pi}{(2-\lambda)\delta}} + B_{3km}\right)\right]\cdot g_{3k,4ik}(u,m,k) \tag{61}$$

$$\begin{aligned}A_{3k} + B_{3k}\ln R_2 \quad &= \frac{1}{(2-\lambda)\delta}\Big[A_{4ik} + B_{4ik}\ln R_2\\ &+ \sum_{v=1}^{\infty}\left(A_{4iku} + B_{4iku}\left(\frac{r}{R_2}\right)^{-\frac{u\pi}{\delta\xi}}\right)\cdot s(u,k)\Big]\end{aligned} \tag{62}$$

$$\begin{aligned}A_{4ik} + B_{4ik}\ln R_2 \quad &= \frac{1}{\delta\xi}\Big[A_{3k} + B_{3k}\ln R_2\\ &+ \sum_{m=1}^{\infty}\left(A_{3km}\left(\frac{R_2}{R_5}\right)^{\frac{m\pi}{(2-\lambda)\delta}} + B_{3km}\right)\cdot r(m,k)\Big]\end{aligned} \tag{63}$$

where

$$l_{3k,4ik}(m,k) = \int_{\theta_k-(\lambda-1)\delta}^{\theta_k+\delta\xi-(\lambda-1)\delta}\cos\left(\frac{m\pi}{(2-\lambda)\delta}(\theta-\theta_k)\right)d\theta$$

$$s_{3k,4ik}(u,k) = \int_{\theta_k}^{\theta_k+(2-\lambda)\delta}\cos\left(\frac{u\pi}{\delta\xi}\cdot(\theta-\theta_k+(\lambda-1)\delta)\right)d\theta$$

$$\begin{aligned}f_{3k,4ik}(u,m,k) \quad &= \int_{\theta_k}^{\theta_k+(2-\lambda)\delta}\cos\left(\frac{m\pi}{(2-\lambda)\delta}(\theta-\theta_k)\right)\\ &\times \cos\frac{u\pi}{\delta\xi}(\theta-\theta_k+(\lambda-1)\delta)d\theta\end{aligned}$$

$$\begin{aligned}g_{3k,4ik}(u,m,k) \quad &= \int_{\theta_k-(\lambda-1)\delta}^{\theta_k+\delta\xi-(\lambda-1)\delta}\cos\left(\frac{m\pi}{(2-\lambda)\delta}(\theta-\theta_k)\right)\\ &\times \cos\frac{u\pi}{\delta\xi}(\theta-\theta_k+(\lambda-1)\delta)d\theta\end{aligned}$$

Then, according to Equation (9), the following equation can be obtained:

$$\begin{aligned}&\sum_{m=1}^{\infty}\left(A_{3km}\left(\frac{R_2}{R_5}\right)^{\frac{m\pi}{(2-\lambda)\delta}} + B_{3km}\right)\cdot l_{3k,4ik}(m,k)\\ &= \sum_{u=1}^{\infty}\left(A_{4iku} + B_{4iku}\left(\frac{R_2}{R_3}\right)^{-\frac{u\pi}{\delta\xi}}\right)\cdot s_{3k,4ik}(u,k)\end{aligned} \tag{64}$$

### 4.4. Interface between Region 4ik and Region 4jk1

According to Equations (18), (19), (21), (23), (35) and (36), the following equation can be obtained:

$$\begin{aligned}B_{4iku} \quad &= 2\Big(A_{4jk1} + A_{4jku1p}\Big)\\ &+ \frac{1}{\delta\xi}\sum_{u=1}^{\infty}\left(A_{4jku1} + B_{4jku1}\left(\frac{R_3}{R_{4jk12}}\right)^{-\frac{u\pi}{\delta}}\right)\cdot l_{4ik,4jk1}(u,k)\end{aligned} \tag{65}$$

$$
\begin{aligned}
A_{4jku1} &= \tfrac{2}{\delta\xi} \cdot \int_{\theta_k}^{\theta_k+(2-\lambda)\delta} \tfrac{\partial A_{4ik}(r,\theta)}{\partial r}\Big|_{r=R_3} \cos\left(\tfrac{u\pi}{\delta\xi}(\theta-\theta_k+(\lambda-1)\delta)\right)d\theta \\
&= \tfrac{2}{\delta\xi R_3} \cdot \left\{ B_{4ik} + \tfrac{u\pi}{\delta\xi}\cdot\sum_{u=1}^{\infty}\left[\left(A_{4iku}\left(\tfrac{R_3}{R_2}\right)^{\tfrac{u\pi}{\delta\xi}}\right) - B_{4iku}\right]\cdot s_{4ik,4jk1}(u,k)\right\}
\end{aligned}
\tag{66}
$$

$$
\begin{aligned}
A_{4ik} + B_{4ik}\ln R_3 &= A_{4jk1} + A_{4jku1p} \\
&\quad + \tfrac{1}{\delta\xi}\cdot\sum_{u=1}^{\infty}\left(A_{4jku1} + B_{4jku1}\left(\tfrac{R_3}{R_{4jk12}}\right)^{-\tfrac{u\pi}{\delta}}\right)\cdot l_{4ik,4jk1}(u,k)
\end{aligned}
\tag{67}
$$

$$
B_{4ik} = \frac{1}{2}\mu_0\left[J_{4jk1}(R_{4jk13}^2 - R_3^2) + J_{4jk2}\left(R_1^2 - R_{4jk13}^2\right)\right]
\tag{68}
$$

$$
\sum_{u=1}^{\infty}\left(A_{4jku1} + B_{4jku1}\left(\frac{R_3}{R_{4jk13}}\right)^{-\tfrac{u\pi}{\delta}}\right) = \sum_{u=1}^{\infty}\left(A_{4iku}\left(\frac{R_3}{R_2}\right)^{\tfrac{u\pi}{\delta\xi}} + B_{4iku}\right)
\tag{69}
$$

where

$$
l_{4ik,4jk1}(u,k) = \int_{\theta_k-(\lambda-1)\delta}^{\theta_k+\delta\xi-(\lambda-1)\delta}\left(1 + \cos\frac{2u\pi}{\delta\xi}(\theta-\theta_k+(\lambda-1)\delta)\right)d\theta
$$

$$
s_{4ik,4jk1}(u,k) = \int_{\theta_k}^{\theta_k+(2-\lambda)\delta}\left(1 + \cos\frac{2u\pi}{\delta\xi}(\theta-\theta_k+(\lambda-1)\delta)\right)d\theta
$$

*4.5. Interface between Region 4jk1 and Region 4jk2*

According to Equations (37)–(41) and (47)–(50), the following equation can be obtained:

$$
\begin{aligned}
A_{4jk2} + \tfrac{1}{2}\mu_0 J_{4jk2}(R_{4jk12}^2\ln R_{4jk12} - \tfrac{1}{2}R_{4jk12}{}^2) &= \\
A_{4jk1} + \tfrac{1}{2}\mu_0 J_{4jk1}(R_{4jk12}^2\ln R_{4jk12} - \tfrac{1}{2}R_{4jk12}{}^2) & \\
+\tfrac{1}{2}\mu_0 J_{4jk2}\left(R_1^2 - R_{4jk12}^2\right)\ln R_{4jk12} &= A_{4ik} + B_{4ik}\ln R_3
\end{aligned}
\tag{70}
$$

$$
B_{4jku2} = A_{4jku2}\left(\frac{R_1}{R_{4jk13}}\right)^{\tfrac{u\pi}{\delta}}
\tag{71}
$$

$$
\sum_{u=1}^{\infty}\left(A_{4jku1}\left(\frac{R_{4jk13}}{R_3}\right)^{\tfrac{u\pi}{\delta}} + B_{4jku1}\right) = \sum_{u=1}^{\infty}\left[A_{4jku2}\left(1 + \left(\frac{R_1}{R_{4jk13}}\right)^{\tfrac{2u\pi}{\delta}}\right)\right]
\tag{72}
$$

by rewriting Equations (51), (52), and (54)–(72) in matrix and vector form. Then, the integral constants of each region can be obtained through matrix operation. In addition, the detailed derivation of $l_{2,3k}$, $s_{2,3k}$, $\gamma_{2,3k}$, $f_{2,3k}$, $g_{2,3k}$, $l_{3k,4ik}$, $s_{3k,4ik}$, $f_{3k,4ik}$, $g_{3k,4ik}$, $l_{4ik,4jk1}$, $s_{4ik,4jk1}$ and the method of matrix operation can be seen in [26].

## 5. FEM Results and Analysis

To verify the proposed analytical model, two LHPMSM prototype machines are manufactured with a semi-closed slot and asymmetric slot, respectively. The FEA models of the two prototypes are shown in Figure 7. The main geometric dimensions are given in Figure 4 and Table 1 and machine parameters are shown in Table 2. They have the same structural design parameters except for the stator teeth. The PMs and stator materials are N38SH and DW470-50, respectively.

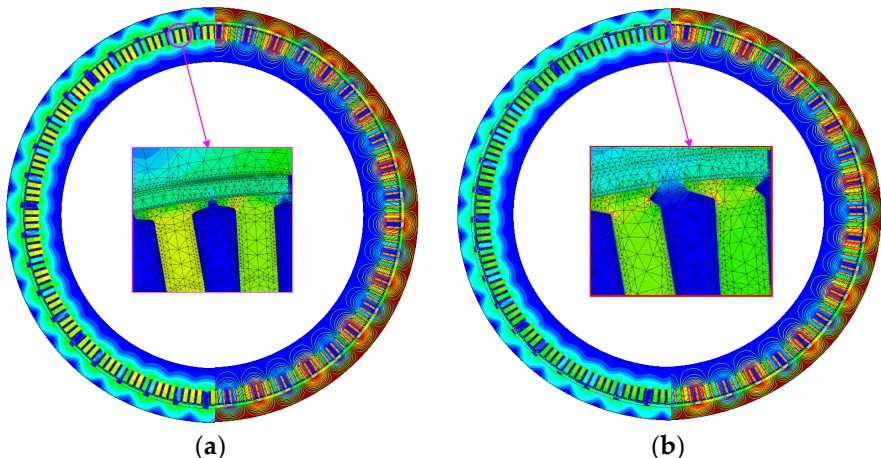

**Figure 7.** The FEM analysis of two LHPMSM prototype machines: (**a**) semi-closed slot; (**b**) asymmetric slot.

**Table 1.** Main Geometric Dimensions of Prototype Machines.

| Symbol | Quantity | Title 3 | |
|---|---|---|---|
| | | Semi-Closed Slot | Asymmetric Slot |
| $p$ | Pole pairs number | 20 | 20 |
| $Q$ | Slot number | 168 | 168 |
| $R_1$ | Radius of slot bottom | 764 mm | 764 mm |
| $R_2$ | Inner radius of slot opening | 824 mm | 824 mm |
| $R_3$ | Outer radius of slot body | 821 mm | \ |
| $R_4$ | Inner radius of PMs | 828 mm | 828 mm |
| $R_5$ | Outer radius of slot opening | 825 mm | 825 mm |
| $R_6$ | Outer radius of PMs | 840 mm | 840 mm |
| $b$ | Tooth width of stator | 20 mm | 20 mm |
| $h_0$ | Toothed boots height | 1 mm | 1 mm |
| $l$ | The depth of U-shaped groove | 1.29 mm | \ |
| $hs_1$ | The width of U-shaped groove | 3 mm | \ |
| $hs_2$ | Stator tooth height | 57 mm | 57 mm |
| $g$ | Air-gap length | 3 mm | 3 mm |
| $\alpha_p$ | Pole-arc coefficient | 0.94 | 0.94 |
| $\lambda$ | The asymmetry of slot opening | variable | 1 |
| $\zeta$ | The asymmetry of inside slot | variable | 1 |
| $\delta$ | Slot pitch angle of stator | 2.14° | 2.14° |

**Table 2.** Parameters of Prototype Machines.

| Symbol | Quantity | Value |
|---|---|---|
| $U_N$ | Rated voltage | 380 V |
| $I_N$ | Rated current | 20 A |
| $P_N$ | Rated power | 12 kW |
| $n_N$ | Rated rotating speed | 50 rpm |
| $R_N$ | Resistance per phase | 1.05 Ω |
| \ | Number of conductors | 28 |
| \ | Number of parallel branches | 8 |

## 5.1. Air-Gap Magnetic Flux Density

Figure 8 shows the comparison between the AM and FEM results of the radial and tangential flux density with the semi-closed slot and asymmetric slot at the middle of the air-gap subdomain (*r* = 826.5 mm). From the figures, although the end effect is ignored, the air-gap flux density waveforms obtained by the two methods are still in good agreement. Moreover, the slotting effect leads to the distortion of the flux density waveforms, while the waveform with the asymmetric slot has higher peaks than that of the semi-closed slot and may have a higher harmonic distortion rate at these locations. Consequently, the harmonic analyses of the air-gap flux density for the two models are shown in Figure 9. The fundamental wave amplitudes of air-gap flux density between the semi-closed slot

model and asymmetric slot model are 1.118T and 1.146T, respectively. In addition, the total harmonic distortion (THD) rate of the air-gap flux density for the two models can be calculated by (73), which are 26.95% and 29.56%, respectively.

$$THD_{ag} = \frac{\sqrt{\sum\limits_{i=1}^{\infty} B_{2i+1}^2}}{B_1} \times 100\% \tag{73}$$

where $B_1$ is the fundamental value of air gap flux density and $B_{2i+1}$ is the $(2i + 1)$th harmonic contents of air gap flux density.

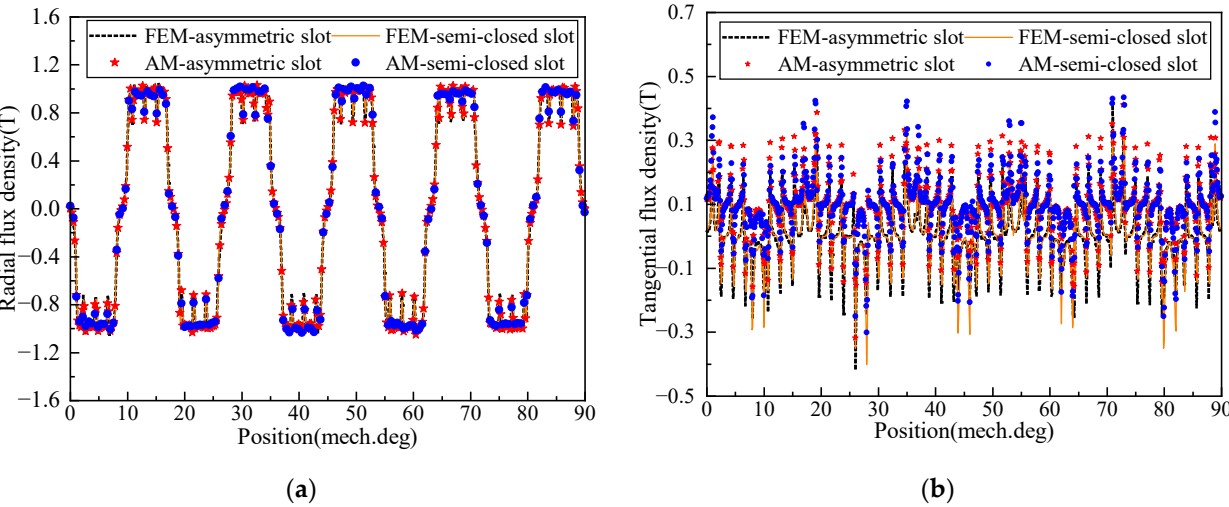

(**a**)                                        (**b**)

**Figure 8.** Air gap flux density of semi-closed slot model and asymmetric slot model: (**a**) radial; (**b**) tangential.

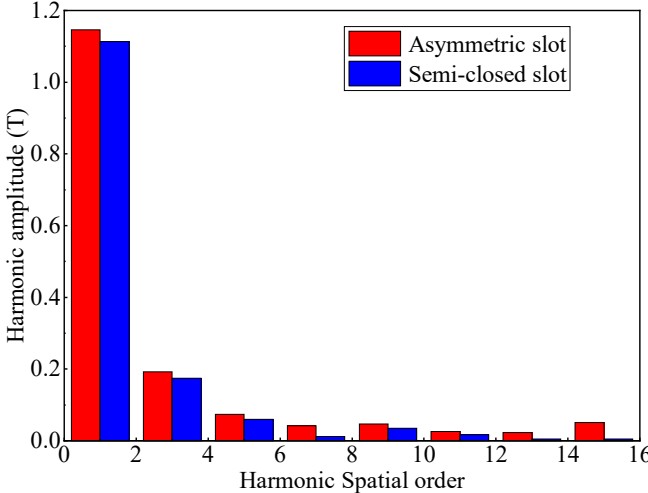

**Figure 9.** Comparison of fundamental wave amplitude of air gap flux density between semi-closed slot model and asymmetric slot model.

As shown in Figure 9, the asymmetric slot model has a higher fundamental wave amplitude of air-gap magnetic flux density than the semi-closed slot model, which indicates that it can withstand a greater magnetic load. This will reduce the size of the motor and, thus, the manufacturing cost. However, the position of the asymmetric slot is slightly different from that of the semi-closed slot in spatial location, which cannot avoid the influence of the slotting effect. Therefore, the asymmetric slot model has a slightly higher

THD of air-gap flux density than the semi-closed slot model, which is mainly caused by the third harmonic content. The harmonic amplitude of the air-gap flux density can be effectively suppressed by optimizing the structure of the stator slot, which will be discussed in Section 6.

Figure 10a shows the 2D-FEM results of no-load back EMF for comparison between the semi-closed slot model and the asymmetric slot model. It can be seen that the amplitudes of the two models—the former is 42.717 V and the latter is 41.677 V—are very close, but the waveform of the latter tends to be a sinusoidal wave. In addition, the harmonic analysis results are shown in Figure 10b. The THD of no-load back EMF for the asymmetric slot model and semi-closed slot model are 11.82% and 10.86%, respectively.

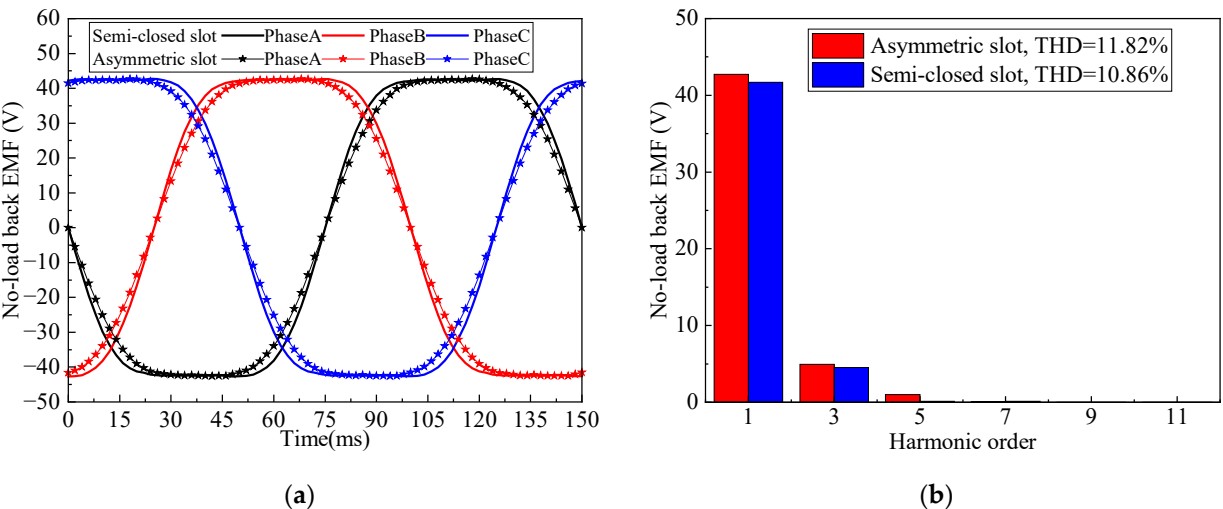

**Figure 10.** Comparison of no-load back EMF between semi-closed slot model and asymmetric slot model: (**a**) waveform; (**b**) harmonic analysis.

### 5.2. Electromagnetic Torque and Cogging Torque

The LHPMSM constantly suffers heavy load and occasionally shock load during operation. Therefore, the driving system requires a high output capacity and stability, which depends on the amplitude of the electromagnetic torque and the torque ripple ratio. Moreover, the cogging torque can cause torque ripple, particularly at low speeds. Consequently, the electromagnetic torque performance and cogging torque performance are analyzed, respectively.

Figure 11a shows the FEM results of the electromagnetic torque of the asymmetric slot model and semi-closed slot model. The average electromagnetic torques of the two models are 2.25 kN·m and 2.19 kN·m, respectively. The corresponding torque ripples of the two models are 2.18% and 2.43%, respectively.

Figure 11b shows the AM and FEM analysis of the cogging torque of the asymmetric slot model and semi-closed slot model. The peak values of the cogging torque are 10.67 N·m and 12.65 N·m, respectively.

Comparisons of electromagnetic characteristics between the asymmetric slot model and traditional semi-closed slot model are shown in Table 3. Note that the '↑' in the table represents the increase rate and the '↓' represents the decrease rate. In summary, the asymmetric slot model reveals better torque performance than the semi-closed slot model, which is mainly because the former can be equivalent to the partially unequal width teeth. This structure can revise the difference of synchronous inductance between two adjacent stator teeth, but the actual effect is affected by the stator tooth structure [21]. Therefore, the cogging torque and torque ripple can be weakened but cannot be eliminated. The torque characteristics will be significantly improved by optimizing the U-shaped groove.

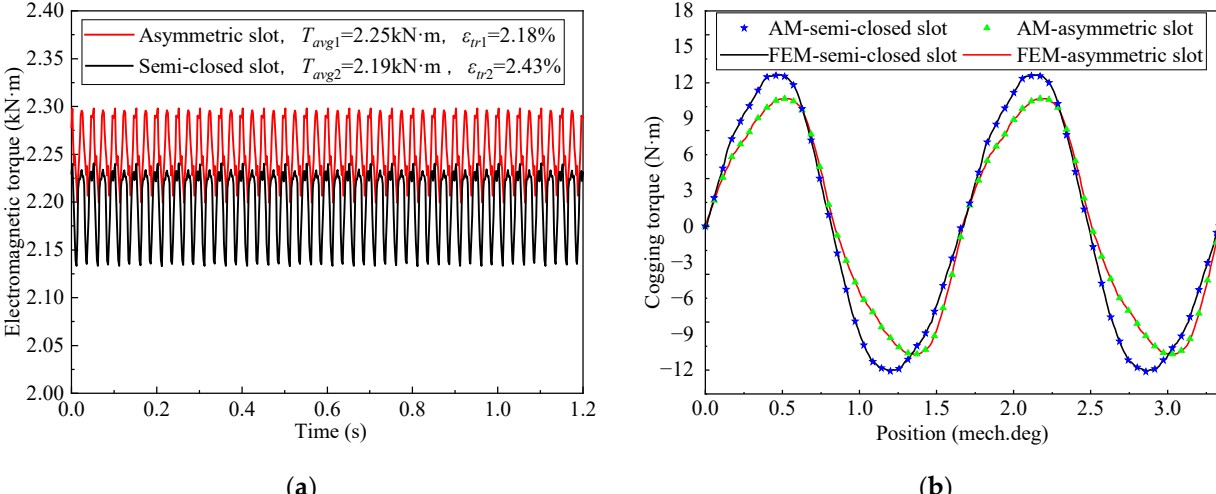

**Figure 11.** Comparison of electromagnetic characteristics between semi-closed slot model and asymmetric slot model: (**a**) electromagnetic torque; (**b**) cogging torque.

**Table 3.** Comparison of Electromagnetic Characteristics Between Asymmetric Slot Model and Traditional Semi-closed Slot Model.

| Comparison | Air-Gap Flux Density (T) | | Air-Gap Flux Density THD (%) | Average Value of Electromagnetic Torque $T_{avg}$ (kN·m) | Torque Ripple (%) | Cogging Torque (N·m) | No-Load Back EMF THD (%) |
| --- | --- | --- | --- | --- | --- | --- | --- |
| | $B_r$ | $B_t$ | | | | | |
| semi-closed slot | 1.118 | 0.076 | 26.95% | 2.19 | 2.43% | 12.65 | 10.86% |
| Asymmetric slot | 1.146 | 0.082 | 29.56% | 2.25 | 2.18% | 10.67 | 11.82% |
| change | 2.4%↑ | 7.3% ↑ | 8.8% ↓ | 2.6% ↑ | 10.2% ↓ | 15.65% ↓ | 8.12% ↑ |

## 6. Influence of Key Parameters

Many motor design parameters, such as the air-gap length, pole–slot ratio, pole-arc coefficient, etc., have a significant impact on electromagnetic characteristics, especially for the air-gap flux density, electromagnetic torque, back EMF, etc. As shown in Section 2, the critical design parameters are $\lambda$ and $\xi$ against the proposed asymmetric slot model. Therefore, the influence of key parameters on electromagnetic performances are investigated in this section.

Based on the relationship between the $\lambda$ and $\beta$ given in Equation (1), the asymmetric slot model with different values of $\lambda$ is modeled by 2-D FEA parametric modeling technology. It should be noted that the feasibility of assembly needs to be taken into consideration. The actual width of the slot opening is only 3 mm for $\lambda = 1.75$. It is difficult to complete wire-wrapping for $\lambda > 1.75$. Thus, the range of $\lambda$ is defined as 1.5 to 1.75, and the step size is 0.05.

Figure 12a shows the influence of $\lambda$ and $\xi$ on the air-gap flux density of the motor. The amplitude of the air-gap flux density increases first then decreases significantly with the increase in $\lambda$ and $\xi$. The air-gap flux density reaches the maximum value of 1.181T when $\lambda = 1.6$ and $\xi = 1.5$. According to Equations (1) and (2), and Table 1, the values of $\beta$ and $\alpha$ are 0.856° and 1.07°, respectively.

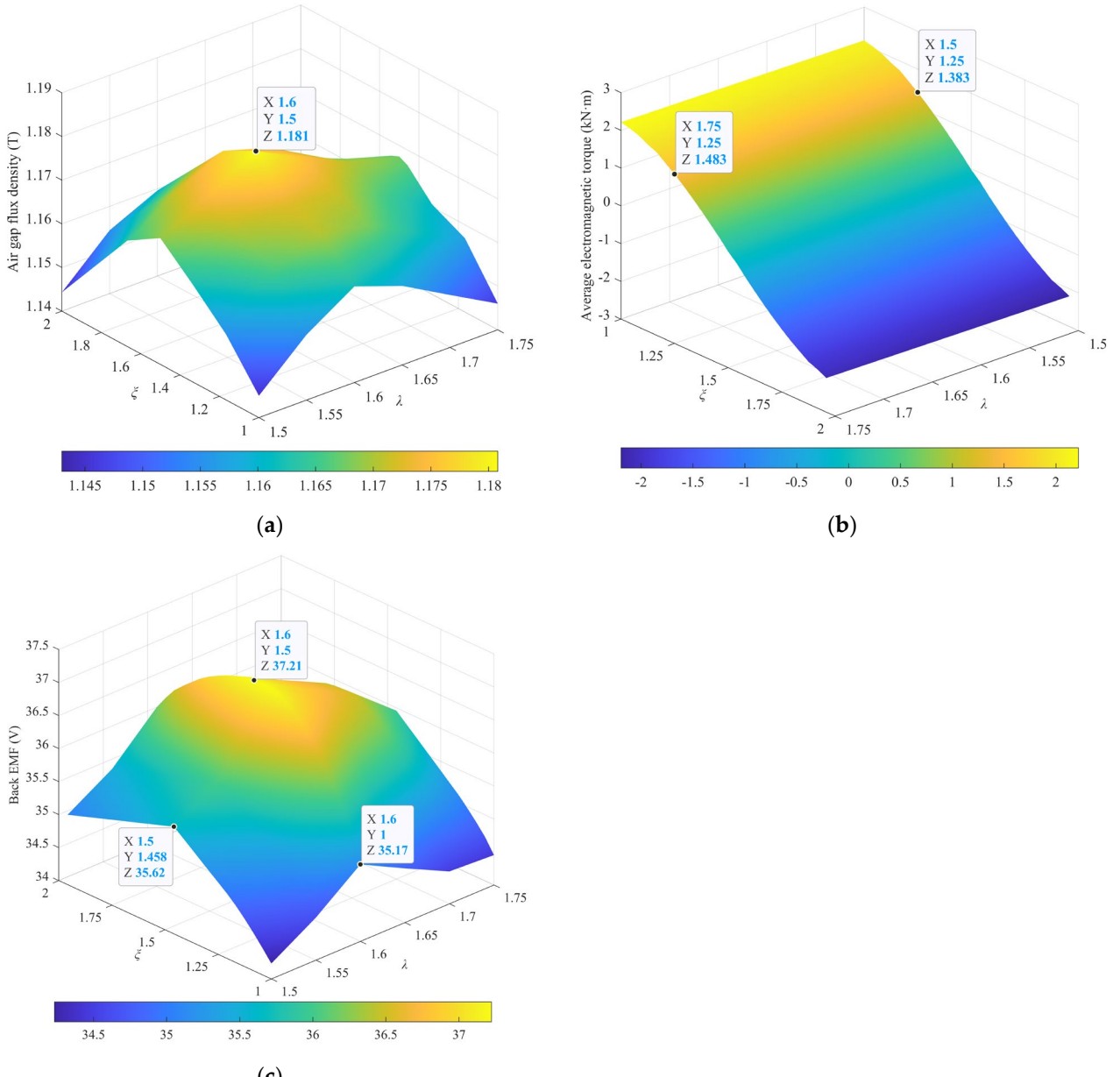

**Figure 12.** Surface chart of electromagnetic characteristics with $\lambda$ and $\xi$: (**a**) fundamental wave amplitude of air-gap flux density; (**b**) electromagnetic torque; (**c**) fundamental wave amplitude of no-load back EMF.

In Figure 12b, as the value of $\lambda$ continues to increase, the average electromagnetic torque is decreased monotonically. Moreover, the direction of electromagnetic torque will be changed when the $\lambda$ reaches a specific value, which is because $\lambda$ is related to the initial position of the slot. On the other hand, as the value of $\xi$ continues to increase, the electromagnetic torque changes slightly.

As shown in Figure 12c, the fundamental wave amplitude of no-load back EMF increases first then decreases significantly with the increase in $\lambda$ and $\xi$. The amplitude of no-load back EMF reaches the maximum value of 37.21 V when $\lambda = 1.6$ and $\xi = 1.5$.

## 7. Experimental Result

To verify the motor performance, an LHPMSM with a 40-pole rotor and 168-slot stator is designed based on the design parameters in Table 1. The values of $\lambda$ and $\xi$ are 1.65 and 1.5, respectively. The experimental platform is shown in Figure 13, and it mainly includes

the experimental prototype, console, hydraulic station, power cabinet, three-level frequency converter, power analyzer, current transformer, and encoder. The frequency converter is set to the LOCAL model, and the data are transmitted to the industrial control computer through the Bluetooth interface.

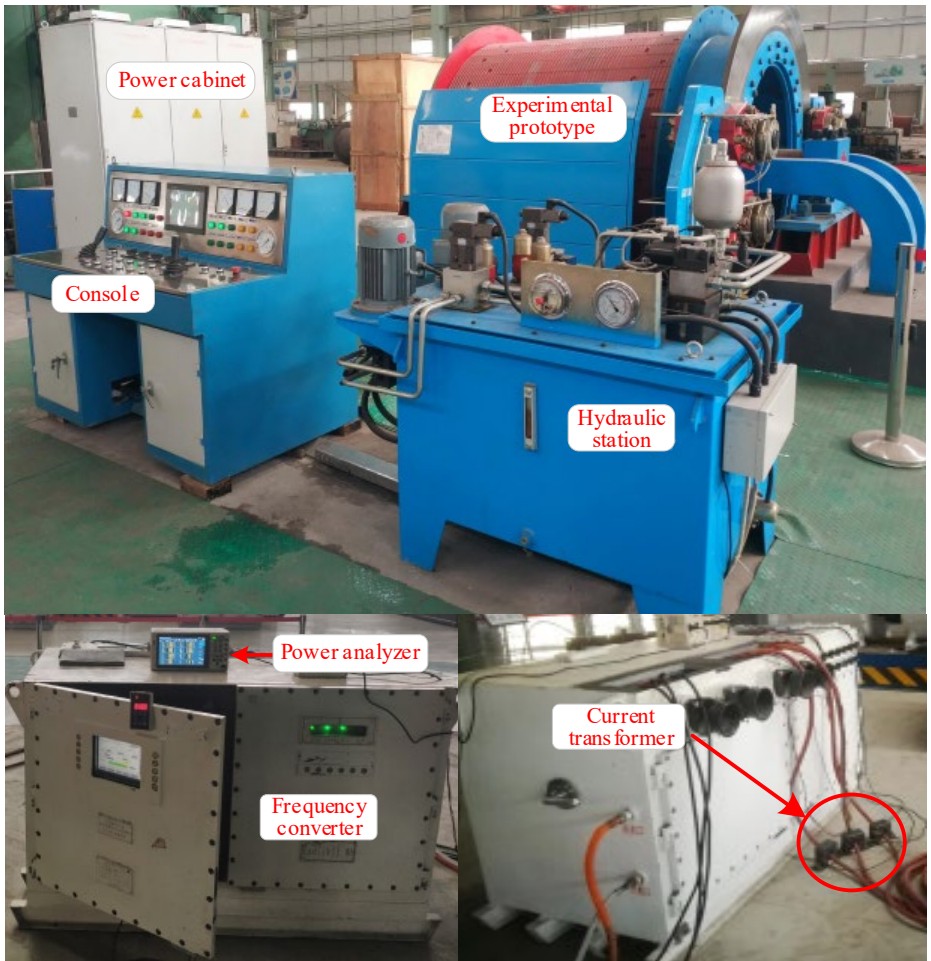

**Figure 13.** The testing platform of the LHPMSM prototype.

As seen from Figure 14, the external surface of the drum is not wrapped with steel wire rope, and its normal operation can be equivalent to the no-load working condition. The experimental prototype rated parameters are set by the three-level frequency converter, as given in Table 2.

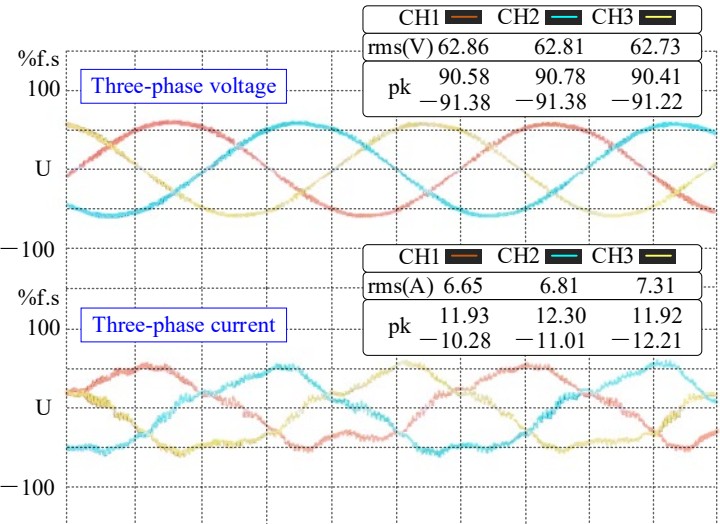

**Figure 14.** Measured three-phase currents and voltages at the rated condition.

The input current and input voltage are measured at 17.2 Hz. The measured waveforms are shown in Figure 15a. The RMS values of the three-phase input voltage are 62.86 V, 62.81 V, and 62.73 V, respectively. The RMS values of the three-phase input current are 6.65 A, 6.81 A, and 7.31 A, respectively.

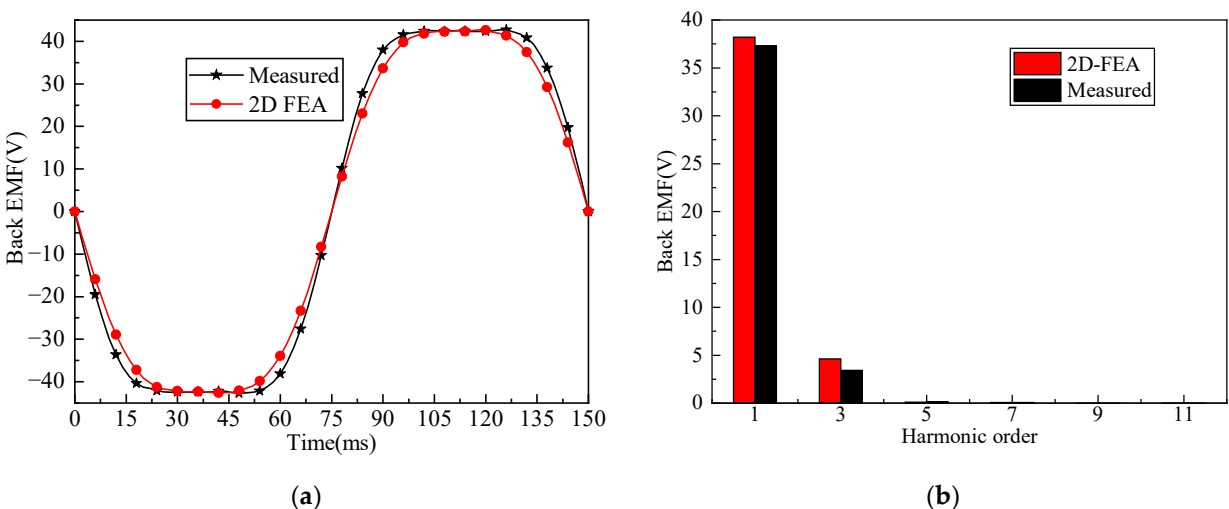

(**a**)  (**b**)

**Figure 15.** Two-dimensional FEM calculated and measured back EMF waveforms at 50 rpm: (**a**) waveforms; (**b**) harmonic analysis.

Figure 15b shows that the measured back EMF is around 2.4% lower than that of the 2D FEM result. It can be calculated that the THD of the 2D FEM and the measured back EMF are 12.09% and 9.16%, respectively.

## 8. Conclusions

In this paper, a novel design of an integrated LHPMSM that includes asymmetric stator slots is proposed. To predict the magnetic field distribution of the complex structures, an analysis model with two correction factors is established and solved by the method of separating variables. The analysis results are in good agreement with the finite element calculation results. In addition, the influential factors of the motor electromagnetic performance are further analyzed. To validate the effectiveness of theoretical analysis, a 168-slot/40-pole LHPMSM with an asymmetric stator slot is manufactured and measured. The following conclusions can be obtained:

(1) For LHPMSM with multiple slots, the effective amplitude of the radial air gap flux density is increased by 2.4%, the harmonic of the air gap flux density is weakened by 8.8%, the average torque of the motor is increased by 2.6%, the torque ripple is weakened by 10.2%, and the cogging torque is weakened by 15.65% compared with the traditional semi-closed stator slot.

(2) The two correction factors, i.e., the asymmetry of the slot opening $\lambda$ and the asymmetry of the inside slot $\xi$ have significant effects on the electromagnetic characteristic of the motor. Specifically, the 168-slot/40-pole LHPMSM has the best electromagnetic characteristics when $\lambda = 1.6$ and $\xi = 1.5$. The values of $\lambda$ and $\xi$ are related to the slot pitch angle of the stator. In addition, we found that the electromagnetic torque is slightly affected by $\xi$. Therefore, it is necessary to properly select the two factors to obtain the optimum electromagnetic characteristic.

**Author Contributions:** S.K.: Simulation, Writing review and editing, Visualization. Z.K.: Conceptualization, Supervision, Methodology, Funding acquisition, Project administration; J.W.: Funding acquisition, Project administration; Y.W.: Formal analysis. All authors have read and agreed to the published version of the manuscript.

**Funding:** National key R&D program-funded projects (grant no. 2020YFB1314103), the National Natural Science Foundation of China (grant no. 52274156), Guizhou science and technology support plan projects (grant no. Guizhou science and technology cooperation support [2022] general 004), Shanxi province raised funds to support the scientific research projects of the returned overseas person (grant no. 2021-049).

**Data Availability Statement:** Not applicable.

**Conflicts of Interest:** The authors declare that they have no known competing financial interest or personal relationships that could have appeared to influence the work reported in this paper.

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
