# Peer review of "Modeling and Simulation of a Novel Low-Speed High-Torque Permanent Magnet Synchronous Motor with Asymmetric Stator Slots"

_machines, doi:10.3390/machines10121143_

Round 1

Reviewer 1 Report

A Modeling and simulation of a novel low-speed high-torque permanent magnet synchronous motor with asymmetric stator slots is proposed. In general, this work can represent an important contribution in modeling and simulation of a synchronous motor, however, there are some observations and questions that must be addressed before a possible publication:

*Authors should include some recent articles to show the relevance and current interest of the proposal.

* The conclusion should be improved by given main contributions, unique findings and numerical results of this manuscript, instead of a general statement.

Author Response

Response to Reviewer 1 Comments

A Modeling and simulation of a novel low-speed high-torque permanent magnet synchronous motor with asymmetric stator slots is proposed. In general, this work can represent an important contribution in modeling and simulation of a synchronous motor, however, there are some observations and questions that must be addressed before a possible publication:

Point 1: Authors should include some recent articles to show the relevance and current interest of the proposal.

Response 1: The authors are grateful to the reviewer for this comment. We agree with the reviewer. We have added 5 relevant references in recent two years to the introduction, and briefly described the research contents and conclusions related to this article in the revised version.

The details are as follows:

  • Literature [1-3]mainly describe the application of low-speed and high torque permanent magnet synchronous motor (LHPMSM) in the industrial field. We understand that the "recent articles" mentioned by the reviewer should be the papers of last two years. Therefore, in the revised version, we have added two articles on the application of LHPMSM to the mining transportation field.
  • Literature [4-6] describe the two traditional transmission modes: direct-drive permanent magnet transmission system and magnetic gear transmission system. Although these two transmission modes are widely used, they have no correlation with the novel LHPMSM proposed in this paper, and therefore theydo not meet the requirements of reviewers on "relevance". Thus, these three documents have been deleted in the revised version.
  • Literature [7-13] describe the optimization methodsof the electromagnetic characteristics for PMSM by designing and optimizing permanent magnet shape and parameters, especially in reducing the cogging torque and the torque ripple, increasing average torque, and weakening air gap flux density harmonics. Some classic structural designs have been proposed in the earlier literature, but the optimization methods are changing with each passing day. As this topic is not the main concern of this paper, in the revised version, we have added two articles on the new segmented inclined method of permanent magnets (PMs) and the design of adjacent unequal pole arc coefficient.
  • Literature [14-20] describe the optimization methods ofthe electromagnetic characteristics for PMSM by designing and optimizing the stator slot shape and parameters, especially in reducing cogging torque. This is the topic of this paper. In the past two years, there have also been many works focusing on the design of new stator slots. However, most of them are still aimed at small or medium-sized permanent magnet synchronous motors with fewer slots. In the revised version, we have added 4 articles, mainly researching on the design, modeling, parameter optimization, and influence on electromagnetic characteristics of the arc-shaped tooth top stator slot, the multi tooth stator slot, the unequal tooth width composite stator slot, and the unequal tooth top width stator slot.

Point 2: The conclusion should be improved by given main contributions, unique findings and numerical results of this manuscript, instead of a general statement.

Response 2: Thank you for your suggestion. We emphasized the main contributions and unique findings of this study in the conclusion of the revised version. In addition, the advantages of the proposed model in quantitative indicators are supplemented.

The revised conclusion is as follows:

In this paper, a novel design of an integrated LHPMSM that includes asymmetric stator slots is proposed. To predict the magnetic field distribution of the complex structures, an analysis model with two correction factors is established and solved by the method of separating variables. The analysis results are in good agreement with the finite element calculation results. In addition, the influential factors of the motor electromagnetic performance are further analyzed. To validate the effectiveness of theoretical analysis, a 168-slot/40-pole LHPMSM with an asymmetric stator slot is manufactured and measured. The following conclusions can be obtained:

1) For LHPMSM with multiple slots, the effective amplitude of the radial air gap flux density is increased by 2.4%, the harmonic of the air gap flux density is weakened by 8.8%, the average torque of the motor is increased by 2.6%, the torque ripple is weakened by 10.2%, and the cogging torque is weakened by 15.65% compared with the traditional semi- closed stator slot.

2) The two correction factors, i.e., the asymmetry of the slot-opening λ and the asymmetry of the inside slot ξ, have significant effects on the electromagnetic characteristic of the motor. Specifically, the 168-slot/40-pole LHPMSM has the best electromagnetic characteristics when λ=1.6 and ξ=1.5. The values of λ and ξ are related to the slot pitch angle of the stator. In addition, we found that the electromagnetic torque is slightly affected by ξ. Therefore, it is necessary to properly select the two factors to obtain the optimum electromagnetic characteristic.

Reviewer 2 Report

- The article is very difficult to read and understand.

- I have read it over and over again to fully understand the work done in the review phase. It needs prof read.

- The authors gave sufficient basic information about the subject in the introduction, and referred to the references related to the subject.

- They explained their proposed method in detail.

- The proposed method was first verified by simulation and then by experimental study, and the results were given in detail.

Author Response

Response to Reviewer 2 Comments

Point 1: The article is very difficult to read and understand.

 Response 1: The authors are grateful to the reviewer for this comment. In the revised manuscript, we have entrusted MDPI (https://www.mdpi.com/authors/ english) for English language editing. The English Editing Certificate can be seen in the attachment.

Point 2: I have read it over and over again to fully understand the work done in the review phase. It needs prof read.

Response 2: Thank you for your suggestion. We have edited the article for language, grammar, and improved clarity.

Point 3: The authors gave sufficient basic information about the subject in the introduction, and referred to the references related to the subject.

Response 3: The authors are grateful to the reviewer for this comment. Thanks for the recognition of our work.

Point 4: They explained their proposed method in detail.

Response 4: Thank you for your suggestion. Thanks for the recognition of our work.

Point 5: The proposed method was first verified by simulation and then by experimental study, and the results were given in detail.

Response 5: Thanks for the comment. Thanks for the recognition of our work.
